# The Impact of the Chronic Disease Self-Management Program on Health Literacy: A Pre-Post Study Using a Multi-Dimensional Health Literacy Instrument

**DOI:** 10.3390/ijerph17010058

**Published:** 2019-12-19

**Authors:** Danielle Marie Muscat, Wenbo Song, Erin Cvejic, Jie Hua Cecilia Ting, Joanne Medlin, Don Nutbeam

**Affiliations:** 1Sydney Health Literacy Lab, Sydney School of Public Health, Faculty of Medicine and Health, University of Sydney, NSW 2006, Australia; erin.cvejic@sydney.edu.au; 2Sydney School of Public Health, Faculty of Medicine and Health, University of Sydney, NSW 2006, Australia; wson6055@uni.sydney.edu.au (W.S.); cecilia.ting@sydney.edu.au (J.H.C.T.); don.nutbeam@sydney.edu.au (D.N.); 3Integrated and Community Health, Western Sydney Local Health District, Blacktown, NSW 2148, Australia; Joanne.Medlin@health.nsw.gov.au

**Keywords:** health literacy, chronic disease, self-management, chronic disease self-management programs

## Abstract

This study assessed the impact of the Chronic Disease Self-Management Program (CDSMP) on different domains of health literacy using a pre-post study design. Participants aged over 16 years and with one or more self-reported chronic diseases were recruited for the CDSMP in western Sydney (a highly diverse area of New South Wales, Australia) between October 2014 and September 2018. Health literacy was assessed pre- and immediately post-intervention using the Health Literacy Questionnaire (HLQ), with differences in mean scores for each HLQ domain analysed using paired sample *t*-tests. A total of 486 participants were recruited into the CDSMP. Of those, 316 (65.0%) completed both pre- and post-intervention surveys and were included in the analysis. The median age of the participants was 68 years, the majority were female (62.5%), and most were born in a country other than Australia (80.6%). There were statistically significant (*P* < 0.001) improvements across all nine domains of the HLQ. This is the first study evaluating the potential impact of the CDSMP on improving different domains of health literacy amongst a diverse sample of participants with chronic diseases using a multi-dimensional instrument. The absence of a control population in this study warrants caution when interpreting the results.

## 1. Introduction

Non-communicable (chronic) disease is a global health problem. Chronic diseases (e.g., cardiovascular diseases, cancer, diabetes, chronic respiratory diseases) are the leading cause of disease burden and mortality worldwide, with the attributable share of total global burden of disease increasing steadily from 44% in 1990 to 61% in 2016 [1]. In Australia, chronic disease accounts for 84–88% of the total disease burden [2]. 

In the absence of a cure, chronic disease must be managed over time as it evolves with shifting severity, pace, and treatments [3]. Holman has written eloquently about the features of chronic illnesses, emphasising their dynamic and evolving nature and variations in severity over time, which require competent, often daily decision making and management by patients [3]. Given this, developing patients’ self-management skills and confidence has become an increasingly important part of clinical care, and a number of interventions aimed at delivering self-management support and education have now been developed and evaluated [4,5]. 

Most chronic disease self-management programs have the educational goal of providing individuals with a range of transferable health-related skills. They recognise that by developing these adaptable skills—for example in symptom recognition, treatment management, and health services use—individuals with established chronic diseases are much better equipped to partner in the management of their disease and respond to the dynamic and evolving nature of their chronic condition. Knowledge improvement is seen as necessary but not sufficient to achieve sustained disease self-management. However, outcomes reported in evaluations of chronic disease self-management programs do not always reflect the skills-directed methods and learning theories that underpin them. Although there is heterogeneity in the outcomes measured in trials of intervention efficacy, a systematic review identified that the most frequently assessed outcomes of chronic disease self-management programs were improved knowledge and self-efficacy, healthcare utilisation and clinical outcomes (e.g., anxiety, depression, energy/fatigue, general health, pain) [6]. 

Health literacy refers to the cognitive and social skills required to gain access to, understand and use information in ways which promote and maintain good health [7]. Although health literacy has been recognised as an important foundation for successful and sustained self-management [8,9,10], few studies have explored the direct impact of chronic disease self-management programs on health literacy skills. In 2013, a pre-post longitudinal evaluation of the Chronic Disease Self-Management Program among middle-aged and older adults was conducted across 17 states in the United States. This study reported that the program led to a statistically significant improvement in health literacy, quantified in this study using a limited measure assessing participants’ confidence filling out medical forms [11].

Health literacy is better understood as a multidimensional construct, moving from a narrow conceptual focus on patients’ skills in reading and writing to also include knowing when and where to seek health information, verbal communication and assertiveness, and skills to retain, process and apply health information in changing circumstances [12]. As understanding and definitions of health literacy have evolved, measurement instruments have been developed to better capture the full breadth of ideas embodied by the concept [13,14]. This has led to the development of more comprehensive measures of health literacy, including the Health Literacy Questionnaire (HLQ) [13] and European Health Literacy Survey [14]. These instruments assess participants’ self-reported capacity to actively manage their health, engage with healthcare providers, navigate the healthcare system and find good health information, among other capabilities. 

The health literacy ‘domains’ captured by these more comprehensive instruments can be matched to the content of self-management interventions which teach participants how to actively manage their health, provide participants with health information and include topics related to communication skills, making decisions and working with health professionals and organisations. The evolution of these more comprehensive measures of health literacy present an opportunity to examine the impact of existing chronic disease management programs on transferable health skills necessary for adaptive and sustainable self-management.

This study aimed to examine the potential impact of the Chronic Disease Self-Management Program (CDSMP) on different domains of health literacy using a pre-post study design.

## 2. Materials and Methods 

### 2.1. Setting 

Western Sydney is a highly diverse area of Sydney, New South Wales (NSW) in Australia. The Western Sydney Local Health District (WSLHD) is one of 15 local health districts (LHDs) in the NSW health system and provides public healthcare to more than one million residents across four Local Government Areas in the west of Sydney. WSLHD has the highest urban indigenous population in Australia, 47% of residents are overseas-born, and one in two speak a language other than English at home [15]. Since 2013, WSLHD has hosted a demonstration program to improve the integration of care between primary and tertiary health systems. One important part of this program was to better support the large group in the population who were living with (often complex) chronic disease, and who were frequent users of the healthcare system. This included making the established CDSMP (described below) available for eligible participants across the health district [16].

### 2.2. Intervention

The CDSMP was developed at Stanford University and has been broadly disseminated across populations and several countries [17,18,19,20,21,22]. The CDSMP comprises structured small-group interventions (2.5 hours each) over six weeks and an accompanying reference book. The program includes aspects encountered throughout the chronic illness trajectory (e.g., fatigue, medication/symptom management, decision-making, communication with providers, and behavioural changes related to nutrition and exercise) [23], and seeks to build self-efficacy for self-management through the mastery of skills (action planning and feedback), symptom reinterpretation, modelling of self-management behaviours, problem-solving strategies, and social persuasion [24] (see Figure 1). The CDSMP is peer-led by trained individuals with a personal experience living with a chronic condition [23].

The content of the CDSMP maps onto the domains of health literacy measured in the HLQ. It teaches participants, for example, how to actively manage their health (HLQ Domain 3) and provides them with information to do so (Domain 2; Domain 8; Domain 9). Topics related to communication skills, making decisions and working with health professionals and organisations may help participants to actively engage with healthcare providers (Domain 6) and feel supported and understood by them (Domain 1), as well as navigate the healthcare system more effectively (Domain 7). The small-group format of the program may also increase perceived social support for health (Domain 4). 

The effectiveness of the CDSMP has been demonstrated in studies across age groups and diverse cultural and ethnic backgrounds [23], including improvements in self-efficacy [17,18,25,26,27], health behaviour (i.e., exercise frequency, symptom management, communication with healthcare providers) [27,28,29], health status (i.e., level of self-reported health, health distress, disability, fatigue, quality of life) [25,28,29,30,31], and health service utilisation (reduction in emergency presentations and hospitalisations) [24,29,31]. Qualitative evaluations suggest that CDSMP is acceptable to participants, and the group-based modality helps enhance participation, social connectedness, coping, and the acceptance of chronic condition(s) amongst participants [20,32,33,34].

In this study, the CDSMP program was delivered by trained bilingual and Indigenous community leaders recruited from multicultural health services, local non-government organisations and volunteer networks. To qualify, all leaders completed 24 h of training across four full-day sessions hosted by master trainers. A total of 27 leaders were trained between 2014 and 2016. Each CDSMP session was led by two trained leaders who followed a detailed *Chronic Disease Self-Management Leader’s Manual* and *Fidelity Manual* to deliver the program. Programs were delivered in several languages (e.g., Mandarin; Cantonese; Turkish; Tagalong; see Table 1) via ‘on-the-spot’ translation by the group leader. All participants were provided with *Living a Healthy Life with Chronic Conditions* reference text, either in English or simplified Chinese. 

### 2.3. Participants and Recruitment 

We used multiple recruitment strategies, with a non-random selection of the study participants. Individuals were referred to the CDSMP through the WSLHD’s multicultural health services, existing chronic disease services (e.g., Connecting Care; Chronic Care Rehabilitation Service), and via clinicians within the Local Health District and general practice network. The inclusion criteria were individuals who (a) had one or more (self-reported) chronic diseases, (b) were over 16 years of age, and (c) had the (self-reported) ability to attend the 6-week face-to-face CDSMP.

### 2.4. Outcome Measures 

The participants were asked to complete paper-based questionnaires providing demographic information and assessing health literacy across a range of domains using the HLQ, as described below. The data were collected upon commencement (pre-intervention) and at the conclusion (post-intervention) of the CDSMP. Although facilitators were present and could assist in reading difficult words, participants were required to answer questions independently.

Health literacy was measured using the HLQ [13]; a multi-dimensional tool that measures health literacy across 9 distinct conceptual domains (Figure 2) [13,35]. The HLQ has 44 questions, with 4 to 5 questions per domain. Domains 1 to 5 assess the *strength of participants’ agreement with a statement* on a 4-point ordinal scale (strongly disagree = 1 to strongly agree = 4) while domains 6 to 9 assess *participants’ perceived ease in task completion* on a 5-point ordinal scale (cannot do or always difficult = 1 to very easy = 5). The median reliability of the HLQ domains is 0.88, with a range from 0.77 to 0.90 [13]. 

The questionnaire takes between 7 to 40 minutes to complete. Participants enrolled in non-English CDSMP groups in this study completed a translated HLQ which were available from the instrument developers.

### 2.5. Statistical Analysis 

Descriptive statistics were used to summarise demographic characteristics and the pre- and post-intervention HLQ scores. The pre-intervention demographic data and mean HLQ domain scores between those who completed the program and those who did not complete the program were compared using chi-square tests and independent sample *t*-tests (see Table A1). Paired sample *t*-tests were used to assess whether the HLQ domain mean scores differed pre- and post-intervention. Exploratory multivariable linear regression modelling was utilised to investigate the associations between the HLQ domain score changes and a range of demographic variables. 

All data were analysed using IBM SPSS Version 25 and R version 3.5.2, with *P* < 0.05 considered statistically significant. The expectation maximisation algorithm was used to substitute the missing HLQ values, following the HLQ User Manual; up to two missing values were imputed for domains with four or five questions. Missing data that could not be imputed were excluded pairwise from the analysis. 

### 2.6. Ethics 

The approval to evaluate and publish the results of this program was obtained from the Westmead Scientific Advisory QA Committee and the Secretary of the WSLHD Human Research Ethics Committee (Protocol number: 1809-04 QA).

## 3. Results

### 3.1. Demographic Characteristics 

In total, 486 participants were recruited for the CDSMP between October 2014 and September 2018. Of those, 38 were excluded from the analysis as they did not complete the pre-intervention questionnaire. An additional 132 participants did not complete the post-intervention questionnaire or were removed from the analysis due to indistinguishable pre- and post-intervention questionnaires, leaving a sample of 316 (65.0%) participants included in the analysis (see Figure 3). 

The pre-intervention demographic characteristics of participants who completed both the pre- and post-intervention questionnaires are summarised in Table 1. Overall, the median age of participants was 68 years. There was approximately twice as many female participants than male participants. The majority were non-native English speakers, did not live alone, had a healthcare card, and were born in Australia, the Greater China regions or India. Health Care Cards are issued by the Australian Department of Human Services to people who live in Australia and already receive government payments, benefits or allowances. Approximately 15% of participants indicated that they required assistance to complete the questionnaires.

Compared to the participants who did not complete the program, those who did were less likely to speak English at home (*P* < 0.001), and more had private health insurance (*P* = 0.04). There were also differences related to country of birth (*P* < 0.001) and teaching groups (*P* < 0.001). Other demographic characteristics were statistically similar (See Table A1). The participants who did not complete the program also had significantly higher mean scores on two health literacy domains pre-intervention compared to those who completed the program (see Table A2): Domain 7 (Navigating the healthcare system) and Domain 8 (Ability to find good health information). However, these groups did not differ significantly on the other seven domains.

### 3.2. Health Literacy

The participants’ mean pre- and post-intervention HLQ domain scores and the mean domain score changes are presented in Table 2. There were statistically significant (*P* < 0.001) improvements in mean scores in all nine domains of the HLQ. The effect sizes (Cohen’s d) ranged between 0.32 (Domain 4: Social support for health) and 0.58 (Domain 2: Having sufficient information to manage my health).

Exploratory multivariable linear regression modelling was conducted (see Table A3) to determine whether the magnitude of change in the HLQ domain scores differed as a function of sociodemographic and health factors. Each HLQ domain score was modelled individually, with all demographic variables shown in Table 1 (except country of birth, due to collinearity with teaching group) included as explanatory variables. The outcomes of these analyses indicated a differential effect of age, gender, living alone, education level, particular chronic health conditions (arthritis, heart problems), and teaching group on the amount of improvement observed across HLQ domains. 

## 4. Discussion

This is the first study to evaluate the impact of the CDSMP on health literacy using a multi-dimensional measure. Analyses showed that the CDSMP was effective in improving all domains of health literacy among a broad cross-section of participants. Statistically significant improvements (corresponding to small to moderate effect sizes) in all nine health literacy domains were observed; and these changes in reported confidence and skills can be matched to the educational content of the intervention. The largest effect sizes were observed for the domains ‘Having sufficient information to manage my health’, ‘Ability to find good health information’, ‘Navigating the healthcare system’ and ‘Appraisal of health information’. The smallest effect size was observed for ‘Feeling understood and supported by healthcare providers’. Exploratory analysis indicated that the differences in health literacy scores according to demographic characteristics were domain-specific, providing a complex picture with no clear patterns. However, there were some notable differences, including that the self-reported ability to ‘actively manage health’ decreased with age; people who lived alone had lower scores on ‘feeling understood and supported by healthcare professionals’; and that, in general, increasing education was associated with higher self-reported understanding of health information.

This study adds to the body of evidence supporting the potential effectiveness of self-management programs for patients with chronic disease [17,24,29,30] by showing that programs such as the CDSMP can support the development of sustainable and potentially transferable health literacy skills immediately post-intervention. Importantly, these findings were observed in a highly diverse cohort of traditionally ‘hard-to-reach’ adults and so provide practical insights into intervention delivery and assessment in this context. While we were able to recruit 486 people into the program using multiple recruitment strategies, 35% did not complete the post-intervention questionnaire and were excluded from the analysis, and those who were included had higher HLQ scores on some domains, and were more likely to have private health insurance. They were, however, less likely speak English at home compared to those who were excluded. High attrition is common in health literacy interventions [36] and it has been acknowledged that work is needed to investigate how to recruit and retain ‘hard to reach’ participants throughout trials [36]. Related to this, our finding that we were able to retain people who spoke languages other than English at home provides some support for the use of in-language programs in diverse settings.

The findings of this study should be considered in light of the limitations of a single-arm pre-post study design with non-random selection of participants, including being unable to account for concurrent events or programs that could have contributed to the change in participants’ health literacy during the intervention period and the possibility of recruiting participants who were highly motivated to improve their health. Given the use of convenience sampling, we do not have any information on the total number of eligible participants, and our sample is not representative of the population in WSLHD or those with chronic disease in Australia. Compared to the total population of Western Sydney Local Health District, for example, our sample had disproportionately more females than males. Although, in Australia, females aged 15 years and over are more likely than males to have one or more chronic conditions, the distribution was more heavily skewed in our sample (62% compared to 36%), compared to population data (57% compared to 51%) [37]. The two main countries of birth (other than Australia) for participants in this study, however, did reflect those of the total population in WSLHD [38]. Future studies applying more rigorous study designs (e.g., matched controls, waitlist-randomised control trial) will help to strengthen evidence for the effectiveness of CDSMP on improving health literacy. 

There are also limitations pertaining to measurement instruments used in this study, particularly the use of translated versions of the HLQ. Although the instrument developers note that a “strict protocol is followed for each translation to help ensure each version of the HLQ is linguistically, culturally and psychometrically robust” [39], the validity and reliability of the translated versions of HLQ used in our study have not been reported in a peer-reviewed format. Studies using skills-based (rather than self-report) measures of health literacy may also yield different results. It has recently been argued that self-report measures of health literacy (such as the HLQ) assess participants’ confidence in their capacity to comprehend and use health materials, rather than their actual ability, and that, conceptually, feelings of perceived confidence (i.e. self-efficacy) differ from health literacy (i.e. skills) [39]. In line with this, a recent meta-analysis found that health literacy had a positive impact on the self-care activities of people living with diabetes only for studies which assessed health literacy using self-report measures and not for those which used skills-based instruments [39]. Comparing the effect of the CDSMP on skills-based and self-report measures of health literacy, therefore, represents an important direction for future research.

To continue to address questions about the key mechanisms through which chronic disease self-management programs are effective [40], future research should build on the current study by also assessing intermediate and longer-term impacts and outcomes in addition to the short-term impacts on health literacy. Structural equation modelling could then be used to develop a more comprehensive program logic and evaluate whether short term impacts (e.g., changes in health literacy) mediate intermediate impacts (improved self-management) and longer-term outcomes (e.g., improved clinical indices).

Although there is now evidence regarding the effectiveness of chronic disease self-management programs—including on health literacy skills identified in this study—Harris et al., [41] have detailed some of the challenges related to implementing and sustaining programs such as the CDSMP in practice. They highlight the importance of collaboration between organisations to provide and coordinate self-management programs (e.g., state health services; non-government organisations; primary care providers), as well as the need for referral pathways which are sustained over time and facilitate direct communication [41]. The education of staff on the evidence of the effectiveness of these programs was also identified as an enabler of engagement with self-management support [41]. A model for implementation has been provided (by Western Sydney Local Health District, as described here) which addresses many of the potential threats to sustainability, including through the development of care pathways and shared care plans between primary, community and hospital providers, and the use of community-based care facilitators to assist in care planning, navigation, transitional care between services, and patient education and self-management [16]. Implementation research exploring models such as these represents an important direction for future work in this field.

## 5. Conclusions

This is the first study to evaluate the potential impact of the CDSMP on improving different domains of health literacy amongst a diverse sample of participants with chronic disease using a multi-dimensional measure. The absence of a control population in this study means that some caution should be applied to interpreting the results. 

## Figures and Tables

**Figure 1 ijerph-17-00058-f001:**
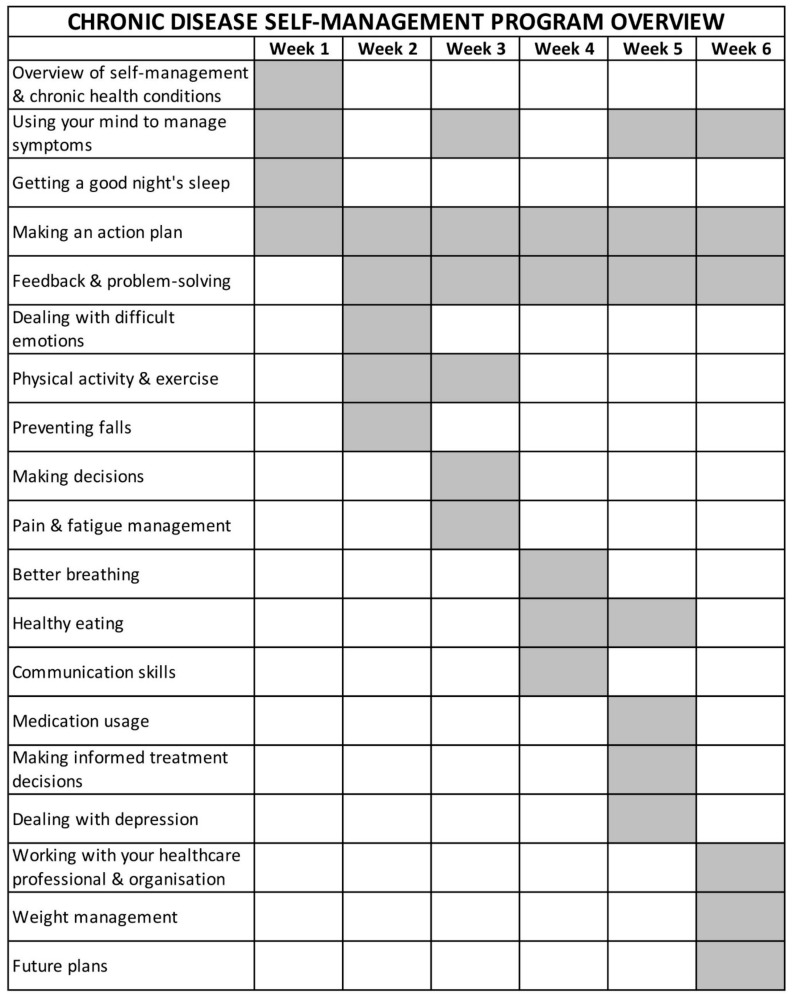
Structure of 6-week Chronic Disease Self-Management Program.

**Figure 2 ijerph-17-00058-f002:**
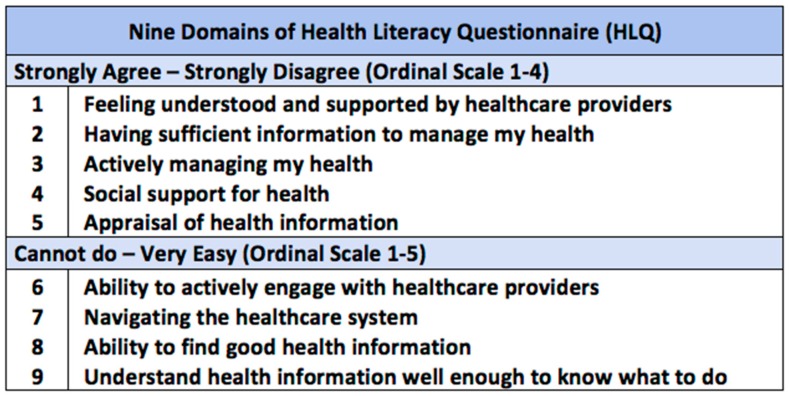
Nine domains of the Health Literacy Questionnaire (HLQ) [13].

**Figure 3 ijerph-17-00058-f003:**
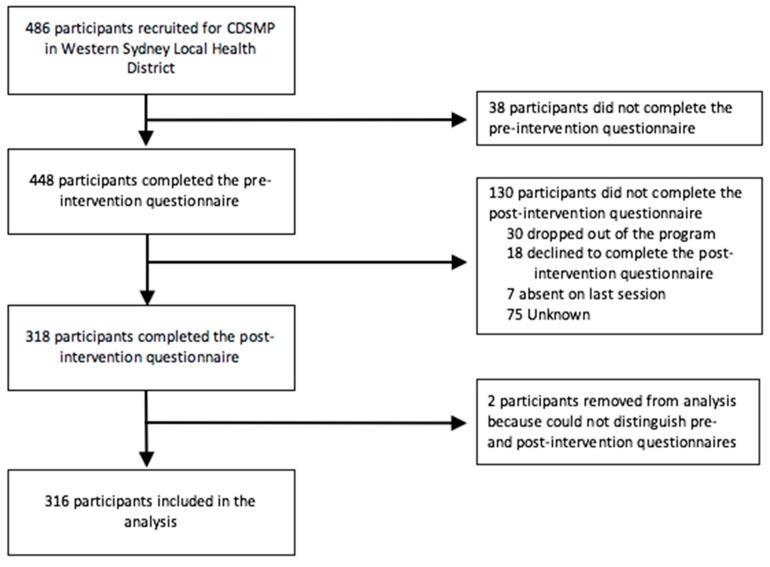
Flow of participants through the study.

**Table 1 ijerph-17-00058-t001:** Demographic characteristics of the participants (pre-intervention) who completed both pre- and post-intervention questionnaires (*n* = 316).

Characteristics	*n* (%)
Age (years)	
<50	12 (3.8%)
50–59	39 (12.3%)
60–69	123 (38.9%)
70–79	104 (32.9%)
≥80	29 (9.1%)
No Response	9 (2.8%)
Gender	
Female	196 (62.0%)
Male	115 (36.4%)
No Response	5 (1.6%)
Live Alone	
Yes	56 (17.7%)
No	235 (74.4%)
No Response	25 (7.9 %)
Country of Birth	
Australia	51 (16.1%)
Mainland China	78 (24.7%)
India	71 (22.5%)
Hong Kong	34 (10.8%)
Turkey	33 (10.4%)
Sri Lanka	25 (7.9%)
Other	19 (6.0 %)
No Response	5 (1.6%)
English Spoken at Home	
Yes	88 (27.8%)
No	222 (70.3%)
No Response	6 (1.9%)
Education Level	
Primary School or Less	32 (10.1%)
High School (not completed)	38 (12.0%)
High School (completed)	66 (20.9%)
TAFE/Trade	42 (13.3%)
Undergraduate	87 (27.5%)
Postgraduate	17 (5.4%)
No Response	34 (10.8%)
Employment	
Working / Studying	14 (4.4%)
Home duties	39 (12.3%)
Retired	148 (46.8%)
Ill / Permanently unable to work	10 (3.2%)
Other	17 (5.4%)
No response	88 (27.8%)
Illness (participants could list more than one condition)	
Arthritis	120 (38.0%)
Diabetes	94 (29.7%)
Back pain	92 (29.1%)
Heart problems	86 (27.2%)
Depression / Anxiety	68 (21.5%)
Other	160 (50.6%)
No response	28 (8.6%)
Private health insurance	
Yes	108 (34.2%)
No	185 (58.5%)
No Response	23 (7.3%)
Have Healthcare Card ^a^	
Yes	230 (72.7%)
No	59 (18.7%)
No Response	27 (8.5%)
Attended Hospital Emergency in the Past 12 Months	
Yes	65 (20.6%)
No	164 (51.9%)
No Response	87 (27.5%)
Required help completing Questionnaire	
Yes	46 (14.6%)
No	247 (78.2%)
No Response	23 (7.3%)
Teaching Group	
Arabic	6 (1.9%)
ATSI	26 (8.2%)
Australian and New Zealander	4 (1.3%)
Chinese (Cantonese)	57 (18.0%)
Chinese (Mandarin)	54 (17.1%)
Chinese (Traditional)	9 (2.8%)
English	22 (7.0%)
Filipino (Tagalong)	1 (0.3%)
Hindi	63 (19.9%)
Indian (English)	12 (3.8%)
Multicultural	4 (1.3%)
Tamil	25 (7.9%)
Turkish	33 (10.4%)

^a^. Health Care Cards are issued by the Australian Department of Human Services to people who live in Australia and already receive government payments, benefits or allowances.

**Table 2 ijerph-17-00058-t002:** Health Literacy Questionnaire (HLQ) Domain Score Changes (*n* = 316). Higher values indicate greater understanding or ability.

HLQ Domain (Range)	Pre-intervention Mean (SD)	Post-intervention Mean (SD)	Mean Score Change (SD)	95%CI	Effect Sizes
1. Feeling understood and supported by healthcare providers (4–16)	11.95 (2.23)	12.62 (1.89)	0.67 (2.16)	(0.43, 0.91)	0.32
2. Having sufficient information to manage my health (4–16)	11.17 (2.31)	12.45 (2.12)	1.28 (2.50)	(1.00, 1.55)	0.58
3. Actively managing my health (4–20)	15.06 (2.53)	16.19 (2.19)	1.13 (2.59)	(0.84, 1.42)	0.48
4. Social support for health (4–20)	14.70 (2.75)	15.53 (2.45)	0.84 (2.81)	(0.52, 1.15)	0.32
5. Appraisal of health information (4–20)	14.51 (2.33)	15.67 (2.29)	1.16 (2.66)	(0.87, 1.46)	0.50
6. Ability to actively engage with healthcare providers (5–25)	17.36 (4.03)	18.79 (3.69)	1.43 (3.76)	(1.01, 1.84)	0.37
7. Navigating the healthcare system (6–30)	19.87 (4.84)	22.22 (4.55)	2.35 (4.68)	(1.83, 2.87)	0.50
8. Ability to find good health information (5–25)	16.39 (4.01)	18.37 (3.82)	1.99 (3.83)	(1.56, 2.41)	0.51
9. Understand health information well enough to know what to do (5–25)	17.16 (3.93)	18.71 (3.93)	1.55 (3.73)	(1.14, 1.96)	0.39

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
