# Peer review of "The Impact of the Chronic Disease Self-Management Program on Health Literacy: A Pre-Post Study Using a Multi-Dimensional Health Literacy Instrument"

_ijerph, 2019, doi:10.3390/ijerph17010058_

Round 1
Reviewer 1 Report
This manuscript is excellent. Thank you for allowing me to read and review as it was a pleasure. The topic of health literacy as it pertains to chronic disease provides greater insight into the need for health education at a global level.

Reviewer 2 Report
This is a well designed study that has relevance internationally due to the topic and sample. The manuscript is well written and logical in flow. The manuscript includes a clear introduction and aim, and reports a well designed study method and results. The discussion section connects study findings to the extant literature and authors report limitations of their study and propose future directions for research based on study findings. I do not have any edits to provide to make this stronger. Thank you for the opportunity to review this high quality manuscript.
Reviewer 3 Report
Thank you for preparing this manuscript for consideration and for the important contribution to existing knowledge about the Chronic Disease Self-Management Program on health literacy.
I have a number of points for your consideration to hopefully improve the clarity and importance of the work.
Section 2. Material and Methods.
Line 147-148. “Participants enrolled into non-English CDSMP groups in this study completed a translated HLQ”. How translation process was conducted and what about validity and reliability of those translated versions of HLQ? By the way, I did not find any information about the reliability of HLQ.
Line 156-157. “Exploratory multivariable linear regression modelling was utilized…”. Not sure about the data of regression analysis.
Section 3. Results.
Line 169-170. Why did the investigation last from 2014 till 2018? Whether it is related to the formation of study groups, participants sampling or other reasons?
Not very clear participants range by age. Table 1. Age < 50 group formed 12 participants. Was it appropriate to form such age group?
Line 212. “Participants who did not complete the program also had higher mean scores on all health literacy domains pre-intervention compared to those who completed the program”. Interesting point for discussion. Perhaps these participants realized that such a intervention would not bring any benefit for them?
Section 4. Discussion.
Line 268, 277. Once again, not very clear sampling. Could you describe briefly “multiple recruitment strategy” used in this study. Or it was non-random selection of participants?
Some general point related with discussion. You are focusing quite a lot on differences of participants who completed and did not completed questionnaires. Also quite a lot discussion regarding limitation of this study. What is ok. But I missed some comments regarding CDSMP, some suggestion for it implementation etc.
